# Neonatal Sequential Organ Failure Assessment (nSOFA) Score within 72 Hours after Birth Reliably Predicts Mortality and Serious Morbidity in Very Preterm Infants

**DOI:** 10.3390/diagnostics12061342

**Published:** 2022-05-28

**Authors:** Ivan Berka, Peter Korček, Jan Janota, Zbyněk Straňák

**Affiliations:** 1Institute for the Care of Mother and Child, 14700 Prague, Czech Republic; peter.korcek@upmd.eu (P.K.); zbynek.stranak@upmd.eu (Z.S.); 2Third Faculty of Medicine, Charles University, 10000 Prague, Czech Republic; 3Neonatal Unit, Department of Obstetrics and Gynecology, Second Faculty of Medicine, Motol University Hospital, Charles University—Prague, V Uvalu 84, 15000 Prague, Czech Republic; jan.janota@fnmotol.cz; 4Institute of Pathological Physiology, First Faculty of Medicine, Charles University, U Nemocnice 5, 12853 Prague, Czech Republic; 5Department of Neonatology, Thomayer Hospital Prague, Videnska 800, 14059 Prague, Czech Republic

**Keywords:** organ dysfunction score, preterm birth, neonatal intensive care

## Abstract

The aim of this study was to assess the applicability of the neonatal sequential organ failure assessment score (nSOFA) within 72 h after delivery as a predictor for mortality and adverse outcome in very preterm neonates. Inborn neonates <32 weeks of gestation were evaluated. The nSOFA scores were calculated from medical records in the first 72 h after birth and the peak value was used for analysis. Death or composite morbidity at hospital discharge defined the adverse outcome. Composite morbidity consisted of chronic lung disease, intraventricular haemorrhage ≥grade III, periventricular leukomalacia and necrotizing enterocolitis. Among 423 enrolled infants (median birth weight 1070 g, median gestational age 29 weeks), 27 died and 91 developed composite morbidity. Death or composite morbidity was associated with organ dysfunction as assessed by nSOFA, systemic inflammatory response, and low birthweight. The score >2 was associated with OR 2.5 (CI 1.39–4.64, *p* = 0.002) for the adverse outcome. Area under the curve of ROC was 0.795 (95% CI = 0.763–0.827). The use of nSOFA seems to be reasonable for predicting mortality and morbidity in very preterm infants. It constitutes a suitable basis to measure the severity of organ dysfunction regardless of the cause.

## 1. Introduction

Neonatal disease severity assessment is an important topic in the neonatal intensive care unit (NICU). In particular, evaluating severity of organ dysfunction among very low birth weight infants, whose average mortality is 11% in developed countries, remains a challenging issue [1] A number of scoring systems have been developed to predict mortality and severe morbidity. Updated Clinical Risk Index for Babies (CRIB II) and the simplified version of the Score for Neonatal Acute Physiology Perinatal Extension (SNAPPE II) are reported to be the most commonly used [2]. However, these schemes are based on static variables combined with biochemical parameters at admission, or several clinical variables shortly after birth [3,4]. In addition, their ability to predict mortality seems to be suboptimal according to recent cohorts [2]. Blood pressure, oxygen saturation, heart rate, and respiratory rate represent changing variables included in the large data models used in the machine learning approach [2,5].

Furthermore, the need for a consensual definition of neonatal sepsis has led to the recent development of a sequential organ failure assessment score (SOFA) which can be applied to neonates and is based on dynamic variables [6]. The neonatal SOFA (nSOFA) has been shown to predict the mortality of very preterm infants with late onset sepsis confirmed by blood culture findings [7]. In the recent large multicenter cohort of preterm patients with necrotizing enterocolitis (NEC), increased nSOFA scores predicted death or need for surgery [8]. Although other morbidities (early onset sepsis) may be more complex for organ dysfunction assessment, nSOFA can provide a standardized scoring algorithm related to mortality and/or morbidity in preterm infants [6,7]. Meanwhile, regular organ dysfunction assessment scoring related to mortality and/or morbidity is not entirely acknowledged in neonatal intensive care units [7]. 

The aim of this study was to evaluate the applicability of organ dysfunction assessment by the nSOFA within 72 h after delivery as a predictor of adverse outcome in very preterm neonates. In these patients, not only death but also long-term serious developmental disorders are of great concern. Major morbidities strongly associated with developmental disorders are chronic lung disease (CLD), peri/intraventricular haemorrhage (PIVH), periventricular leukomalacia (PVL), and necrotizing enterocolitis (NEC). We hypothesized that elevated nSOFA values representing respiratory and cardiovascular instability could predict a composite outcome of death or major morbidity. Given the nature of this scoring system developed for the conditions of presumed sepsis, the relationship between systemic inflammation or early onset sepsis (EOS) and organ dysfunction assessed by nSOFA was also studied.

## 2. Materials and Methods

### 2.1. Patients and Data Collection

This retrospective cohort study was conducted in the single tertiary NICU of the Institute for the Care of mother and Child in Prague, Czech Republic. Data from the period of January 2015 to December 2017 were analysed. An overview of patients born <32 weeks of gestation in the target period was created in the hospital information system. Inclusion criteria were: preterm delivery <32 weeks of gestation, inborn neonates and completed blood count and inflammatory markers measurements within the first 72 h after the delivery as a part of the standard care. Patients with congenital malformations and those who were not resuscitated at birth were excluded. Data were collected from medical records and a laboratory information system. Subsequently, patients with incomplete medical records and an unknown hospital outcome (discharge <36 weeks) were also excluded. Sample size was calculated based on the proportion of patients with the adverse outcome using alpha 0.05. The required sample size was 310. The study protocol was reviewed by local Ethics Committee. Approval was not required due to the retrospective nature of the study. However, written informed consent for anonymous use of clinical data was obtained from parents of each infant.

### 2.2. Blood Sampling

Blood samples were taken from indwelling arterial or venous catheters in all infants within the first 72 h after delivery. Complete blood count, including leukocyte and platelet counts, interleukin (IL)-6 and C-reactive protein (CRP) were routinely measured. The highest level of inflammatory biomarkers was considered the peak value. Blood cultures with a volume of at least 1 mL were collected after birth or when early onset sepsis (EOS) was suspected. The BacT/Alert automated blood culture monitoring system (BacT/Alert, BioMerieux, Durham, North Carolina, USA) was used. Blood counts were measured with a Coulter Micro Dif II (Coulter Electronics Ltd., Fullerton, CA, USA). IL-6 level was assessed by electrochemiluminescence immunoassay (Cobas 6000, e601 module, Roche Diagnostics, Mannheim, Germany). CRP was measured by immunoturbidimetry (Cobas 6000, c501 module, Roche Diagnostics, Mannheim, Germany).

### 2.3. Definitions

Hospital outcome was either survival without severe morbidity, with severe morbidity, or death. Severe morbidity was defined as a composite of chronic lung disease (CLD), intraventricular haemorrhage ≥grade III according to Papile classification, presence of periventricular leukomalacia (PVL) on cranial ultrasound in the postmenstrual age of 36 weeks and necrotizing enterocolitis (NEC) ≥IIb according to modified Bell criteria. CLD was defined as need for oxygen and/or respiratory support at 36 weeks of postmenstrual age [9]. Death or major morbidity defined adverse hospital outcome.

Early systemic inflammation was classified according to serum IL-6 and CRP peak values. IL-6 values > 200 ng/L and CRP > 10 mg/L within 72 h after delivery represented a systemic inflammatory response [10].

The definition of EOS was based on positive blood culture and clinical signs of infection. Clinical signs of early onset infection included: hypothermia, respiratory instability (apnea, desaturations, respiratory distress syndrome with ongoing mechanical ventilation), cardiac (cyanosis, bradycardia, poor peripheral perfusion, hypotension) and neurological (lethargy, suspected seizures) symptoms [11]. Suspected EOS was defined by clinical signs of infection and antibiotic treatment over 48 h despite negative blood culture.

### 2.4. The nSOFA Score

The nSOFA system uses categorical scores containing respiratory, cardiovascular and hematologic components. It comprises presence of invasive ventilatory support, oxygen requirements to maintain normal peripheral saturation, inotropic/vasoactive treatment, corticosteroids administration and the presence and severity of thrombocytopenia [6] (Table 1). 

The nSOFA score was calculated from medical records after admission to the NICU throughout first 72 h after birth. Calculator available online was used (http://www.peds.ufl.edu/apps/nsofa/default.aspx, accessed on 1 February 2021). For the purposes of this analysis, the maximum score values and their time after birth were recorded. For zero score values, the time of admission from the delivery room wasrecorded.

### 2.5. Respiratory and Circulatory Support

Respiratory distress syndrome (RDS) was prevented and treated as recommended by the European Consensus Guidelines on the Management of Respiratory Distress Syndrome [12,13]. In patients with spontaneous breathing efforts nasal continuous positive airway pressure (NCPAP) was used for delivery room stabilisation. Surfactant replacement therapy was provided according to early rescue protocol. NCPAP failure was defined by the fraction of inspired oxygen (FiO_2_) above 0.3 in infants ≤26 weeks of gestation and above 0.4 in infants >26 weeks of gestation [12,13]. In patients with ventilatory support or oxygen therapy born <32 weeks of gestation, the standard oxygen saturation range was 87–95%. An extubation attempt was considered at a mean airway pressure of less than 8 mbar and a FiO_2_ of less than 0.4.

Hypotension was defined as mean arterial blood pressure (measured in mmHg) lower than gestational age in weeks. Circulatory failure was defined by hypotension and clinical or laboratory signs of hypoperfusion, such as prolonged capillary refill time of 3 s or more and elevated serum lactate levels (>5 mmol/L). When echocardiography was used, left ventricular output and left ventricular fractional shortening were measured and evaluated. Inotropes or steroids were initiated according to circulatory failure assessment. Dopamine alone or in combination with dobutamine was the treatment of choice. Hydrocortisone therapy was considered when dopamine doses above 15 µg/kg/minute were required [14].

### 2.6. Statistical Analysis

Data analysis was made using the IBM SPSS Statistics 23.0.0.0 software (IBM Corp., Armonk, NY, USA). Patient demographics and clinical characteristics are presented as medians and interquartile ranges for continuous variables and for categorical variables as counts and category percentages. Chi-square, Kruskal-Wallis and Mann-Whitney U tests were used to compare non-normally distributed variables. Two-sided *p*-values < 0.05 were considered statistically significant. Spearman’s rank correlation coefficient (Spearman’s ρ) was used to assess the relationship between inflammatory biomarkers and organ dysfunction scoring due to non-normal data distribution.

The multinomial (multiple) logistic regression with stepwise forward method was utilized to evaluate the probability of the different possible outcomes of dependent variable (adverse hospital outcome as per our definition), given a set of independent (perinatal) variables. The statistical test allowed us to explore more than two independent variables (as opposed to binary logistic regression) with the added benefit of not assuming normality or linearity.

Receiver operating characteristics curve (ROC) was generated to plot true positive rate (sensitivity) against false positive rate (1—specificity) across varying threshold settings. The ROC curve for risk of death or major morbidity development by nSOFA was analysed and diagnostics parameters for identified cut-off values provided. To determine the association of individual morbidities forming the composite outcome and the increased values of the nSOFA score, we compared the relative risks of these morbidities with respect to the identified cut-off value.

## 3. Results

Out of 454 infants eligible for the study 31 were excluded (Figure 1). Four hundred and twenty-three infants were included in the analysis (median birth weight 1070 g, median gestational age 29 weeks). Mortality in the study group was 6.4% (27 patients). Median postnatal age at death was 7 days (IQR 4–18). Fourteen patients (51.9%) died during the first week after birth as a direct consequence of severe cardio-respiratory instability early after birth, six of whom died during the first 72 h. The most common causes of death in the first week after delivery were pulmonary haemorrhage (4 cases), persistent pulmonary hypertension (4 cases) and severe PIVH (3 cases). After the first week, the leading causes of death were late onset sepsis (5 cases) and severe PIVH (3 cases). PIVH deaths included those for whom care objectives were redirected due to a very severe prognosis quod vitam et sanationem. The incidence of major morbidities in the study group was 15.4% (65 patients) for CLD, 7.3% (31 patients) had intraventricular haemorrhage ≥ grade III, 4.7% (20 patients) were diagnosed with PVL, and 4.5% (19 patients) with NEC. Forty-four patients (10.4%) suffered from more than one major morbidity. Overall, composite morbidity was recorded in 91 cases (21.5%).

Demographic data and perinatal variables according to hospital outcome are presented in Table 2. Peak nSOFA values in the first 72 h after birth differed significantly between groups without major morbidity and with adverse outcome (Figure 2). When dividing the group with adverse outcome into deceased and survivors with major morbidity, the nSOFA peak values were again significantly different among groups. The median in the group without major morbidity was 0 (IQR 0–1), in the group with major morbidity 2 (IQR 0–6) and in the group of deaths 11 (IQR 7–14). The p-value of the independent samples Kruskal-Wallis test was <0.001 (Figure 3). The time of the highest score also varied significantly among groups. The median time of the highest score was 3 h after birth in patients without severe morbidity. In non-survivors and patients with severe morbidity was the median time 21 and 24 h after birth respectively. The correlation of the maximum nSOFA score with serum inflammatory biomarkers was low. Spearman’s ρ was 0.482 for CRP (*p* < 0.001) and 0.434 for IL-6 (*p* < 0.001). Correlation of nSOFA and leukocyte count was not found (Spearman’s ρ—0.013).

Only two infants had confirmed EOS (Escherichia coli, Candida albicans), one survived with severe morbidity. However, 51 neonates (12.1%) were treated with antibiotics for more than 48 h despite negative blood culture. The median peak nSOFA scores among patients with confirmed or suspected early onset sepsis were higher in comparison with patients not treated with antibiotics over 48 h (9, IQR 4–12 vs. 0, IQR 0–1, *p* <0.001).

Logistic regression revealed a significant relationship of adverse hospital outcome and birthweight, systemic inflammatory response and the peak nSOFA score within 72 h after birth (Table 3). The nSOFA score cut-off values were determined by the optimal binning procedure in SPSS. Gestational age, SGA and intubation at delivery room were not included in the model due to loss of statistical significance. 

The receiver operating characteristics curve (ROC) for risk of death or major morbidity development by nSOFA had an area under the curve (AUC) of 0.795 (95% CI = 0.763–0.827). The nSOFA > 2 had a sensitivity of 67% and a specificity of 80% for the composite adverse hospital outcome, with a positive predictive value (PPV) of 57% and a negative predictive value (NPV) of 86%. The positive likehood ratio was 3.4 and the accuracy of this nSOFA level was 77%. The nSOFA > 11 had a sensitivity of 23% and a specificity of 99%, with PPV of 93% and NPV of 77%. The positive likehood ratio was 35 and the accuracy of this nSOFA level was 78%. The numbers of patients according to the peak nSOFA and the hospital outcome are provided in Table 4.

The relative risk for nSOFA scores > 2 was greater than one for all individual morbidities of the composite outcome. It was highest in NEC (1.44) and IVH (1.28), lower in CLD (1.2) and PVL (1.1).

## 4. Discussion

The presence of life-threatening organ dysfunction is a reliable predictor of in-hospital mortality and adverse outcomes in adult and pediatric patients primarily with suspected infection [15,16]. The nSOFA score represents an effort to provide organ dysfunction assessment that may be suitable for very preterm infants. The score has been shown to be associated with mortality in patients with late onset sepsis and necrotizing enterocolitis [7,8]. However, there is a lack of information on the possible use of nSOFA in the early neonatal period, especially during the first 72 h after birth. Moreover, early serious complications in very preterm infants leading to organ dysfunction are common but may not be necessarily caused by systemic inflammation or EOS [6]. These are mainly acute cardio-respiratory illnesses (RDS and its complications—pulmonary hypertension, pulmonary haemorrhage; hypotension requiring intervention; hemodynamically significant persistent ductus arteriosus), perinatal asphyxia and IVH [17].

Our study revealed that the nSOFA score during the first 72 h after birth could be a useful tool for detecting an increased risk of mortality and adverse hospital outcome. We identified a cut-off value similar to the previous study by Fleiss et al. [7]. The nSOFA score > 2 was associated with a 2.5-fold increase in the odds of the adverse outcome. The diagnostic properties of this increased score were acceptable, the accuracy was moderate. Of the morbidities that make up the composite outcome, there appeared to be the highest risk of developing NEC and IVH. In the cited study, the non-survivors differed in the progression of organ dysfunction expressed by the respiratory and circulatory components of the score [7]. Our patients with severe morbidity and non-survivors also seemed to be more likely to suffer from progressive organ dysfunction with peak scores approximately 24 h after birth. These findings support routine clinical use of continuous standardized assessment.

The reported ability of frequently used systems to predict neonatal mortality is variable, but very good for SNAPPE II (AUC 0.849–0.91) and markedly different for CRIB II (from 0.667 for mortality cases in ≤7 days and 0.708 >7 days to 0.92) [2]. The prediction of morbidity is not good in the former and absent in the latter [3,4]. Hence, neonates at high risk of death can be already reliably identified by the SNAPPE II score. This system also includes an assessment of cardiorespiratory instability, albeit shortly after birth. It is the continuation of the assessment that could improve the composite risk evaluation in our view.

We found that activation of systemic inflammation is also significantly associated with the adverse outcome, regardless of its etiology. Systemic inflammation as a risk factor for adverse outcome in preterm infants has been reported previously [18,19]. The weak correlation between the nSOFA score and serum inflammatory markers highlights already mentioned diverse origins of organ dysfunction in very preterm infants especially within first 72 h of life. Transitional disorders, particularly acute respiratory illness and circulatory disturbances can be based exclusively on immaturity of organ systems [20]. Thus, cases with early organ dysfunction, systemic inflammation and suspected or microbiologically confirmed early onset infection are partially overlapping [21].

The relationship between nSOFA and EOS could not be analyzed in detail, since the number of confirmed blood stream infections was very low in contrast to a much larger cohort [22]. Negative blood cultures could be caused by insufficient sample volume or maternal antibiotic treatment [23]. Suboptimal sampling may have not been recorded in our study and maternal antibiotics were not analysed. The ratio of suspected to confirmed EOS (25:1) was also very high compared to the highest reported values (16:1) [23]. We detected strong relationship between increased peak nSOFA values and suspected EOS, but a low correlation with serum CRP and IL-6. Therefore, prolonged administration of antibiotics in our study population was presumably based on clinical conditions rather than positivity of inflammatory biomarkers. 

Respiratory management has an obvious effect on nSOFA values. Endotracheal intubation and initiation of mechanical ventilation within postnatal stabilization in very preterm neonates is very variable even among similar European regional centres (30% vs. 70%) [24]. The European average is over 40% [25]. The incidence of 25.5% within our study group can thus be considered low. This value reflects the preference for non-invasive ventilatory support in the immediate postnatal period. The risks of low birth weight and low gestational age for neonatal mortality and morbidity are clearly established. Low birth weight in the logistic regression model combines gestational age and SGA variables. Hence, its importance for the prediction of the adverse outcome was statistically significant. The incidence of SGA in the study population was 12.7%. This is consistent with assessment by neonatal birth weight standards [26]. Our findings confirm that being born SGA is a significant risk factor for mortality and adverse outcome [27]. 

The results of our study underline the value of a sequential scoring system specifically dedicated for very preterm infants. The nSOFA seems to have desirable properties of neonatal illness severity scoring systems emphasized previously: easy to use, applicable early after birth, useful regardless of various categories of preterm neonates and reliable in terms of mortality prediction [28]. Proponents of the nSOFA suggest and implement organ dysfunction scoring into electronic health records of NICU patients to enable prospective studies and quality improvement [7]. The potential benefit is high. Anticipated nSOFA extension and collection of large data could support unified routine organ dysfunction assessment in the NICU.

This study has several limitations. The nSOFA scores could have been influenced by our center guidelines, specifically for initiation of circulatory support. Differences in the threshold for inotropic/vasoactive treatment can significantly affect the nSOFA value. Also, the use of a composite outcome with a wide variability of consequences can be misleading. We therefore tried to provide at least indicative information on the risk of developing individual morbidities if the nSOFA is above the cut-off value. Retinopathy of prematurity (ROP) was not included among the morbidities associated with long-term developmental disorders. The main reason was the influence of local therapeutic protocols on the incidence of severe forms, mainly intravitreal anti-VEGF administration. In addition, there was an extensive overlap between severe ROP (≥3), PIVH and CLD in our patients. We did not compare the ability to predict mortality using nSOFA with other frequently employed scoring systems. SNAPPE II score differs by a limited time interval of application (up to 12 h after delivery), with CRIB II the problem is uncertain current validity and focus only on the prediction of mortality. The comparison could thus be significantly skewed. Unbalanced patient groups and retrospective design represents other limitations to mention.

In summary, ongoing standardized monitoring of the development and progression of organ dysfunction appears to be a reasonable tool for predicting mortality and short-term morbidity in very preterm infants. Furthermore, nSOFA provides a suitable basis for assessing the severity of organ dysfunction regardless of etiology. More specific patterns of organ dysfunctions associated with EOS remain to be elucidated.

## Figures and Tables

**Figure 1 diagnostics-12-01342-f001:**
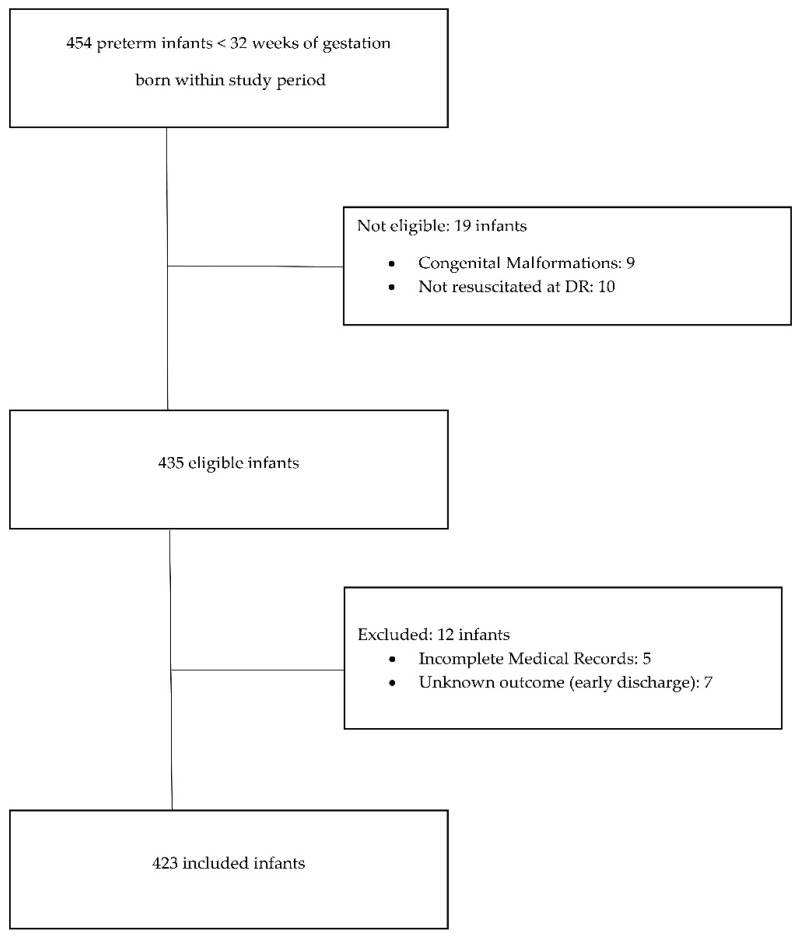
Flow chart. DR = delivery room.

**Figure 2 diagnostics-12-01342-f002:**
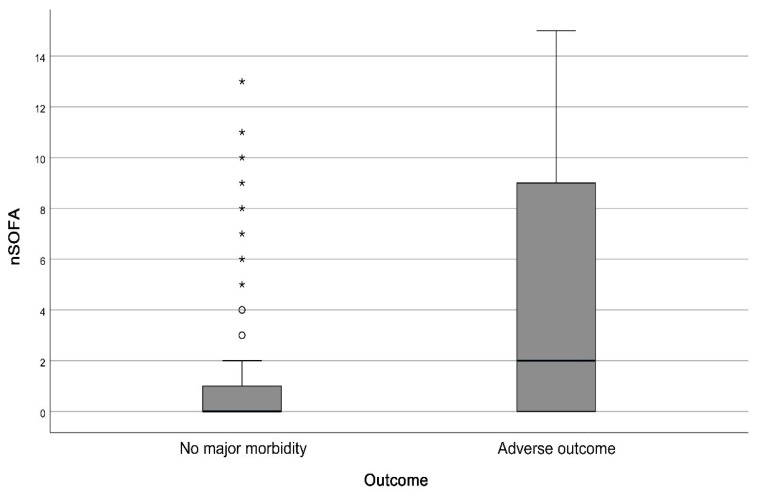
Comparison of nSOFA peak values in the first 72 h after birth between the group with no major morbidity (median 0, IQR 0–1) and with adverse outcome (median 2, IQR 0–9). Mann-Whitney U test, *p*-value < 0.001. Circles represent outliers, stars represent far out values according to Tukey’s fences. nSOFA = Neonatal Sequential Organ Failure Assessment. Adverse outcome = Death or major morbidity.

**Figure 3 diagnostics-12-01342-f003:**
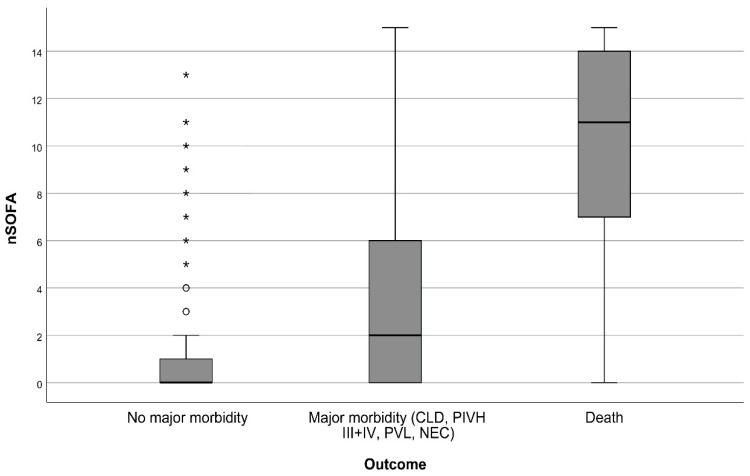
Comparison of nSOFA score peak values in the first 72 h after birth in subgroups without major morbidity (median 0, IQR 0–1), with major morbidity (median 2, IQR 0–6) and death (median 11, IQR 7–14). Independent samples Kruskal-Wallis test, *p*-value < 0.001. Circles represent outliers, stars represent far out values according to Tukey’s fences. nSOFA = Neonatal Sequential Organ Failure Assessment. CLD = Chronic Lung Disease. PIVH = Peri/Intraventricular Haemorrhage. PVL = Periventricular Leukomalacia. NEC = Necrotizing Enterocolitis.

**Table 1 diagnostics-12-01342-t001:** Neonatal sequential organ failure assessment score (nSOFA).

Respiratory Score	0	2	4	6	8
Criteria	Not intubated or intubated, SpO_2_/FiO_2_ ≥ 300	Intubated, SpO_2_/FiO_2_ < 300	Intubated, SpO_2_/FiO_2_ < 200	Intubated, SpO_2_/FiO_2_ < 150	Intubated, SpO_2_/FiO_2_ < 100
Cardiovascular score	0	1	2	3	4
Criteria	No inotropes, no systemic steroids	No inotropes, systemic steroid treatment	One inotrope, no systemic steroids	At least two inotropes or one inotrope and systemic steroids	At least two inotropes and systemic steroids
Hematologic score	0	1	2	3	
Criteria	Platelet count ≥ 150 × 10^9^/L	Platelet count 100–149 × 10^9^/L	Platelet count < 100 × 10^9^/L	Platelet count < 50 × 10^9^/L	

**Table 2 diagnostics-12-01342-t002:** Demographic data of the study population according to hospital outcome. Continuous data are presented as median and interquartile range (IQR), categorical variables are presented as number (percentage).

Variable	No Major Morbidity (N = 305)	Death or Major Morbidity (N = 118)	*p*-Value
Gestational age, weeks Birth weight, grams Female	29 (28–30)	26 (25–28)	<0.001 ^a^
	1160 (950–1430)	750 (620–1040)	<0.001 ^a^
	133 (43.6)	47 (39.8)	0.512 ^b^
SGA * Twins	25 (8.2)	29 (24.6)	<0.001 ^b^
	135 (44.3)	59 (50)	0.482 ^b^
C-section ANS completed Intubation at the delivery room Confirmed or suspected EOS	253 (83)	96 (81.4)	0.775 ^b^
	216 (70.8)	81 (68.6)	0.554 ^b^
	45 (14.8)	63 (53.4)	<0.001 ^b^
	14 (4.6)	39 (33.1)	<0.001 ^b^
Systemic inflammatory response	19 (6.2)	36 (30.5)	<0.001 ^b^
Peak nSOFA	0 (0–1)	2 (0–9)	<0.001 ^a^

Abbreviations: SGA = small for gestational age, ANS = antenatal steroids, EOS = early onset sepsis, nSOFA = neonatal sequential organ failure assessment. ^a^ Mann-Whitney U test. ^b^ Chi-Square. * Birth weight below the 10th percentile.

**Table 3 diagnostics-12-01342-t003:** Multiple logistic regression (stepwise forward) for adverse hospital outcome (death or severe morbidity) of very preterm infants in study population.

		95% CI for OR		
Variable	OR	Lower	Upper	*p*
BW	0.997	0.996	0.998	<0.001
SIR	3.196	1.402	7.282	0.006
nSOFA > 2	2.541	1.392	4.639	0.002
nSOFA > 11	21.183	4.026	111.458	<0.001

Abbreviations: OR = odds ratio, CI = confidence interval, BW = birth weight, SIR = systemic inflammatory response, nSOFA = neonatal sequential organ failure assessment.

**Table 4 diagnostics-12-01342-t004:** Numbers of patients according to peak nSOFA and hospital outcome.

nSOFA Peak Values in the First 72 h	No Major Morbidity	Death or Major Morbidity	Total
0–2	245	39	284
3–11	58	52	110
>11	2	27	29
Total	305	118	423

Abbreviations: nSOFA = Neonatal Sequential Organ Failure Assessment.

## Data Availability

The data that support the findings of this study are available on request from the corresponding author, IB. The data are not publicly available due to their containing information that could compromise the privacy of research participants.

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
