# Peer review of "Neonatal Sequential Organ Failure Assessment (nSOFA) Score within 72 Hours after Birth Reliably Predicts Mortality and Serious Morbidity in Very Preterm Infants"

_diagnostics, 2022, doi:10.3390/diagnostics12061342_

Round 1

Reviewer 1 Report

I have read this paper with great interest, and value the effort of the authors. I only have some additional suggestions to further improve the readability of the paper.

To improve the readability of the paper for the readership, i recommend to either explain somewhat then SOFA score in the abstract or introduction, or at least, put the nSOFA score higher in the methods section ?

Severe morbidity: visual and hearing impairment are not (yet) included in the morbidity model ? was this intentional

What’s the rationale to explore both the nSOFA on outcome and inflammatory biomarkers ? I would assume that ‘inflammatory events’ occur, but are for sure not the main or single driven of neonatal outcome, as modulated by immaturity ? The same holds likely true for leukocytosis, as both hyper- as hypo-leukocytosis are relevant, so that a correlation a priori has its limitations.

Reviewer 2 Report

This manuscript presents the results of a retrospective cohort study seeking to assess the performance of the nSOFA score in predicting death or a composite adverse outcome in infants born at <32 weeks of gestation. The authors discuss that the novelty of using the nSOFA score over CRIB II or SNAPPE II primarily relies in its ability to integrate data past the first few hours after delivery. This is indeed valuable and an important topic of investigation that could potentially aid in early discussions of prognosis with the families of these high-risk infants.

This reviewer has several major concerns regarding the manuscript as written, however, that the authors should consider addressing:

  1. The interpretation of composite outcomes is always tricky, but especially so when the composite consists of endpoints with a large variability in outcomes/consequences. For instance, the long-term outcomes for an infant with severe periventricular hemorrhage is considerably different from one who needed low flow oxygen at 36 weeks corrected age but discharges home on room air - both of these populations are combined under the authors' composite outcome, however. This makes the clinical translatability of these analyses much more difficult. The authors could consider something like a sensitivity analysis where they look at each of the individual outcomes separately to determine if there is one of the diagnoses that is primarily driving the association between nSOFA and the composite outcome and therefore provide the reader with more information regarding which outcome(s) the elevated score may be most associated with.
  2. Although regression analyses are helpful to adjust for possible confounders, ORs are a difficult concept to integrate into clinical counseling. For prognostic markers, it tends to be more helpful to provide metrics such as sensitivity, specificity, NPV, PPV, and perhaps ROC analyses. This seems particularly important given the significant overlap in the box plots in Figs. 2 and 3.
  3. The use of peak values also limits clinical translatability. It significantly limits the potential for interpreting values because clinicians would need to wait until it is clear that the values have peaked before determining an infant's risk. It also reduces the sensitivity of the analyses presented, since it is likely that a baby who peaks at nSOFA=11 in the first 24 hours and then stays at 11 for the next two weeks is a very different baby from one who peaks at 11 in the first 24 hours, gets surfactant, and then goes back down to 1 and stays around there. Did the authors consider assessing patient nSOFA values at certain time points (e.g. 24 hours and 72 hours) instead of using peak values?
  4. The authors mention in the limitations section that they didn't compare to other score systems because they primarily used data in the first 12 hours of life. It is not clear, though, why this precludes a comparison, and it would be a very valuable addition to this study to compare the nSOFA performance to the other well validated scores.

Additional minor concerns:

Abstract/Keywords

  • Line 23: "the adverse outcome" is a vague term. Consider replacing with "composite morbidity" if that is what the authors are referring to
  • Consider using keywords that are not already in the title

Introduction

  • There are some minor grammatical errors throughout the manuscript (e.g. should be "the neonatal intensive..." in line 33, line 36 is missing a period, etc)
  • The authors provide a mortality statistic in line 35, but they are discussing two different populations (VPT and VLBW) in that sentence and it is unclear which of those two populations that statistic applies to
  • The introduction focuses on the ability of previous scores to assess mortality, but does not provide adequate rationale for why the current composite outcome was chosen
  • The second paragraph of the introduction is generally unclear with regard to its goals and main points. Consider rewriting to improve clarity and flow
  • Lines 59-60: "organ dysfunction" does not appear to be defined in the methods and it is unclear whether it is included anywhere in the results or discussion. Either it should be removed, or its meaning clarified in the following sections.

Materials and Methods

  • As it is not a common practice for CBC and inflammatory markers to be obtained on all preterm infants as is described as "standard care" in line 66, it would be helpful if the authors further explain this practice including what is drawn and when, as well as some general information on how they use these data clinically (i.e. is it possible that these labs being elevated biases the care by resulting in different care for those with high values versus lower).
  • What does "unknown outcome" refer to in line 66? Is there a more specific description that could be provided (Fig 1 suggests this is "early discharge" but this is also not clear - is early discharge occurring before 36 weeks)?
  • When were head ultrasounds performed for determination of cPVL?
  • Line 93: did the diagnosis of "systemic inflammation" require elevations of both IL-6 and CRP or should this state "or" instead of "and"?
  • In the respiratory support section, consider reporting oxygen needs as FiO2 instead of %inspired O2 (i.e. 0.3 instead of 30%, etc) in order to be more consistent with the use of FiO2 in the nSOFA score
  • Line 116: Again, is this truly "and" or should it be "or"?
  • Lines 133-134: The sentence starting with "For zero score values..." is unclear. Should it state something like "recorded" instead of "evaluated"?
  • Why were two-sided analyses chosen? Did the authors hypothesize that increased nSOFA scores might be associated with either improved or worsened outcomes?

Results

  • Line 149: which morbidity are the authors referring to when they state "the initial early morbidity"?
  • Figs. 2 and 3 need a description of what the difference is between stars and circles

Discussion

  • Consider adding other common sequelae that are likely not driven by inflammation or EOS to the list on line 218. These could include hypotension, IVH, etc.
  • Consider a different word in line 247, because prolonged administration of antibiotics are never "indicated" without a positive culture, so consider something like "driven by" or "based on"

Reviewer 3 Report

In this study, reliability of nSOFA score for mortality and serious morbidity in very preterm infants was assessed.

The study is well written, however has some concerns that should be addressed before potential publication.
  • Materials and Methods section should be expanded and explained in detail regarding methodology, sample size calculation and patient inclusion
  • Place of study conduction is missing
  • Statistical analysis section should be expanded with detailed explanations of every statistical test used (normality of data distribution, correlation etc.)
  • Authors should remove references from the results - everything that is important for the result interpretation should me mentioned in the Methods section
  • Figures 2 and 3 are somewhat confusing - abbreviations and explanations of the symbols used is missing from the legend

Round 2

Reviewer 2 Report

The authors have addressed most of the reviewers' comments. This reviewer continues to believe that the difference in timing between the author's nSOFA analyses and the previous literature regarding CRIB II and SNAPPE II does not preclude a discussion comparing the predictive value of the different scores. Given that the other two scores make up the preponderance of the current literature, integrating these data into the discussion is important to tie the results of the current study to the previous literature.

Additionally, on line 249, please specify what "them" refers to in "all of them."

Reviewer 3 Report

No further comments.
